# TEMPERATURE-SCALING SURPRISAL ESTIMATES IMPROVE FIT TO HUMAN READING TIMES – BUT DOES IT DO SO FOR THE "RIGHT REASONS"?

**Tong Liu & Iza Škrjanec & Vera Demberg**
Saarland University
Saarbrücken, Germany
`tongliu.physics@gmail.com, {skrjanec,vera}@coli.uni-saarland.de`

## ABSTRACT

A wide body of evidence shows that human language processing difficulty is predicted by the information-theoretic measure *surprisal*, a word's negative log probability in context. However, it is still unclear how to best estimate these probabilities needed for predicting human processing difficulty – while a long-standing belief held that models with lower perplexity would provide more accurate estimates of word predictability, and therefore lead to better reading time predictions, recent work has shown that for very large models, psycholinguistic predictive power decreases. One reason could lie in the fact that language models might be more confident of their predictions than humans, because they have had exposure to several magnitudes more data. In this paper, we test what effect temperature-scaling of large language model (LLM) predictions has on surprisal estimates and their predictive power of reading times of English texts. Firstly, we show that calibration of large language models typically improves with model size, i.e. poorer calibration cannot account for poorer fit to reading times. Secondly, we find that temperature-scaling probabilities lead to a systematically better fit to reading times (up to 89% improvement in delta log likelihood), across several reading time corpora. Finally, we show that this improvement in fit is chiefly driven by words that are composed of multiple subword tokens.[1]

## 1 INTRODUCTION

In psycholinguistics, a key finding is that words with higher surprisal (= negative log probability of the word in context) require more time for processing (Hale, 2001; Levy, 2008). Numerous studies provided experimental evidence supporting this theory, demonstrating that surprisal is a powerful predictive measure of processing complexity (e.g., Demberg & Keller, 2008; Wilcox et al., 2020; 2023; Shain et al., 2022), and that the relationship between surprisal and reading times (RTs) indeed seems to be linear (Smith & Levy, 2013; Wilcox et al., 2020; Shain et al., 2022).

However, prior work implicitly made the assumption that human predictability estimates would be similar to the actual probability of a word occurring in a given context, and that therefore, surprisal values estimated from models that achieve lower perplexities should also approximate human processing difficulty better (Goodkind & Bicknell, 2018; Merkx & Frank, 2021).

Recent research has however found that this is not true – surprisal values from very large LLMs provide in fact a very poor fit to reading times. Oh & Schuler (2023) hypothesize that this might be due to LLMs being "too confident" in their estimates of rare named entities compared to humans, thanks to their manifold larger exposure to data and greater memory capacity compared to humans. Furthermore, work on NLP applications like question answering has reported that probability estimates from pretrained language models are often overconfident, i.e. they are higher than the ground truth probability (Si et al., 2022; Kumar, 2022). These findings hence beg the question whether current LLMs are well-calibrated with respect to "objective" word occurrence probabilities. Relat-

---

[1]Code in this paper will be released upon paper acceptance.

edly, we ask whether LLM probability estimates are overconfident compared to human estimates (as observed in reading times).

One approach to address calibration problems is to use *temperature scaling*, as done e.g., in vision tasks (Guo et al., 2017; Hendrycks et al., 2019). Temperature-scaling with a temperature $T > 1$ has the effect that the probability distribution is flattened such that it becomes more similar to a uniform distribution. Temperature-scaling hence incorporates uncertainty into the probability estimates from LLMs.

We note that the idea to work with flattened distributions instead of the original probability distributions from LLMs is also related to contextual Rényi Entropy as discussed by Pimentel et al. (2023). Pimentel et al. find that Rényi entropy with $\alpha = 0.5$ can explain RTs better than Shannon entropy ($\alpha = 1$). Similarly to temperatures greater than 1, $\alpha$ values lower than 1 have the effect of making the distribution more similar to a uniform distribution. We'd like to point out however that our work and that of Pimentel et al. (2023) differ in the assumed underlying reasons for why a slightly flattened distribution may be more suitable, and whether this change in distribution is applied when calculating surprisal vs. when calculating entropy.

Our experimental results show that scaling probabilities can largely improve the fit to reading times in all 12 settings (3 corpora $\times$ 4 neural LMs). Our contributions are summarized as follows: (1) We propose temperature-scaled surprisal, where surprisal is calculated from temperature-scaled probabilities. (2) We demonstrate that temperature-scaling with temperature T≈2.5 improves predictability of human reading times of English texts compared to T=1. (3) We identify linguistic phenomena that correlate with the benefit of temperature-scaled surprisal by analyzing residual errors from regression models. (4) We relate temperature-scaled surprisal to contextual Rényi entropy.

## 2 PREDICTIVE POWER FOR READING TIMES

In psycholinguistics, RTs on a word are believed to correlate with its processing difficulty. RTs can be gathered using different paradigms, including eye-tracking while reading text on a screen (Rayner, 1998), self-paced reading (Aaronson & Scarborough, 1976; Mitchell & Green, 1978) and the Maze task (Forster et al., 2009).

The most common procedure for predicting words' RT is first to select a set of predictor variables thought to impact RTs $\mathbf{v} = [v^{(1)}, ..., v^{(d)}]^\top \in \mathbb{R}^d$, which include, e.g., the length of a word $w_t$, $|w_t|$, the frequency of a word $\text{freq}(w_t)$. Let $f_\phi : \mathbb{R}^d \to \mathbb{R}$ be a regression model parametrized by $\phi$ used to fit these predictors for the prediction of human RTs $rt$: $rt(w_t|\boldsymbol{w}_{<t}) \sim f_\phi(\mathbf{v})$, given the previous context $\boldsymbol{w}_{<t}$. The performance of $f_\phi$ is quantified by its log-likelihood, with a higher log-likelihood indicating a better psychometric predictive power for human RTs (Frank & Bod, 2011; Fossum & Levy, 2012).

Besides the word length $|w_t|$ and word frequency $\text{freq}(w_t)$, a word's surprisal (i.e., its negative log-probability in context) Hale (2001); Levy (2008) has been shown to be predictive of RTs (Demberg & Keller, 2008; Goodkind & Bicknell, 2018; Wilcox et al., 2020; Shain et al., 2022).

## 3 METHODS

In this section, we delve into key aspects of information-theoretic measures in language comprehension. We start with surprisal, a method connecting processing difficulty to word predictability. As word predictability is empirically estimated by LLMs, we introduce the notion of calibration errors, metrics quantifying how good the estimation of word predictability is. Further, we lay out temperature-scaled surprisal, and the relation between varying temperature vs. varying $\alpha$ in contextual Rényi entropy.

### 3.1 SURPRISAL

Starting from Shannon (1948), the information conveyed by a word $w_t$ has been quantified as the negative log probability of the word $w_t$ given its previous context $\boldsymbol{w}_{<t}$. In Surprisal Theory (Hale, 2001; Levy, 2008), this quantity is called surprisal $s(w_t)$ and proposed to be predictive of the word's

processing difficulty, typically quantified as its RT. Surprisal values are typically estimated from language models $\hat{p}(w_t|\boldsymbol{w}_{<t})$.

$$s(w_t) = -\log_2 p(w_t|\boldsymbol{w}_{<t}), \tag{1}$$

## 3.2 Calibration error

**Definitions** Let $\mathcal{D} = \{(x_i, y_i)\}_i^N$ be a data set where $x_i \in \mathcal{X}$ is an sample (i.e., context) and $y_i \in \mathcal{K} = [K]$ is a category label. Let $g_\theta$ and $\hat{\mathbf{z}}_i = g_\theta(x_i)$ denote a language model parametrized by $\theta$ and the output logit vector of sample $i$, respectively. The predicted class label $\hat{y}_i$ for sample $i$ is given by $\hat{y}_i = \arg\max_{k \in \mathcal{K}} g(x_i)_k$ and confidence for sample $i$ is given by $\hat{p}_i = \max_{k \in \mathcal{K}} g(x_i)_k$. A model is perfectly calibrated when the confidence $\hat{p}$ is equal to the frequency of correctness, i.e., $\mathbb{P}(\hat{y}_i = y_i|\hat{p}_i = p) = p$ holding for all $p \in [0, 1]$ and any sample $i$. Any difference between the left and right sides of the above equation indicates there exists a *calibration error*.

**Expected calibration error (ECE) (Guo et al., 2017)** ECE is the most popular calibration metric, which empirically approximates the calibration error by discretizing the probability interval into a fixed number of bins ($B_m$ with $m \in \{1, 2, ..., M\}$), and measures the gaps of averaged confidence and averaged accuracy in each bin $B_m$.

$$\text{ECE} = \frac{1}{N} \sum_{m=1}^{M} | \sum_{i \in B_m} \hat{p}_i - \sum_{i \in B_m} \mathbb{1}[\hat{y}_i = y_i]|, \tag{2}$$

where $\mathbb{1}$ is the indicator function. However, it does not necessarily measure the actual-word probability, which is the probability required for calculating surprisal in Eq. 1. It focuses only on the top-label probability for a given sample.

**Classwise-ECE (CECE) (Kumar et al., 2019; Kull et al., 2019)** In comparison, CECE measures probabilities of all classes. For each bin and every class $k$, it assesses the difference between the average confidence of samples for class $k$ and the actual proportion of class $k$. If assuming all classes weigh equally, we have:

$$\text{CECE} = \frac{1}{NK} \sum_{k=1}^{K} \sum_{m=1}^{M} | \sum_{i \in B_m} \hat{p}_{i,k} - \sum_{i \in B_m} \mathbb{1}[k = y_i]|, \tag{3}$$

where $\hat{p}_{i,k}$ is the predicted probability of sample $i$ for class $k$.

**Human-likeness calibration error (HCE)** We define the HCE as the Kullback-Leibler divergence (KL divergence) between predicted probability $\hat{\boldsymbol{p}}$ from a neural LM and actual probability $\boldsymbol{p}^*$ of *human* language model.

$$\text{HCE} = D_{KL}(\hat{\boldsymbol{p}}||\boldsymbol{p}^*). \tag{4}$$

Empirically, since $\boldsymbol{p}^*$ is not directly observable, we approximate it by the estimates of a temperature-scaled model that best fits reading times (as discussed later). We denote the approximated HCE using such a method as $\text{HCE}_{\text{TS}}$.

## 3.3 Temperature-scaled surprisal

Temperature scaling (Guo et al., 2017) is a widely-used method to improve model calibration. Given the output logit vector $\hat{\mathbf{z}}_i$ for sample $i$, a single scalar $T > 0$ is applied to rescale $\hat{\mathbf{z}}_i$ before the softmax activation:

$$\hat{q}_i = \max_k \sigma_{SM}(\frac{\hat{\mathbf{z}}_i}{T})^{(k)}, \tag{5}$$

where $\hat{q}_i$ is the calibrated confidence for sample $i$, and $\sigma_{SM}$ is the softmax function. Scaling by a scalar $T$ does not alter the ranking; hence, the predicted label $\hat{y}_i$ remains unchanged. As $T > 1$, it "softens" the probability distribution (i.e., makes the distribution *more uniform*), *increasing uncertainty and entropy* of the probability distribution, while $T < 1$ peaks the distribution. The parameter $T$ in research on calibration is optimized by minimizing the negative log-likelihood on

the validation set. In our experiments of fit to human RTs, we manually tune this temperature with $T > 1$.

Temperature scaling has been successfully applied in several applications: In knowledge distillation (Hinton et al., 2015), temperature scaling (with $T > 1$) is used to "soften" the knowledge (i.e., probability distribution) provided by the teacher model; in text generation, temperature is used to shape the probability distribution to ease certain aspects of the problems of top-k sampling (e.g., choosing an appropriate $k$ value across varying contexts) (Ficler & Goldberg, 2017; Fan et al., 2018). Temperature tuning inherently shifts the model's output in the generation's quality/diversity spectrum (Caccia et al., 2018), with higher temperature decreasing the quality of generation while improving its diversity. This also aligns with our consideration of a possibility that human probability distributions might be flatter than the ones learned by language models and thus increasing the predictive diversity of surprisal provided by LLMs could potentially yield more human-like distributions.

Given Eq. 5, temperature-scaled surprisal is:

$$s_T(w_t, T) = -\log_2(\sigma_{SM}(\hat{\mathbf{z}}_{w_t}/T)^{(k^*)}),\tag{6}$$

where $\hat{\mathbf{z}}_{w_t}$ and $k^* = y_{w_t}$ denote the logit vector and the actual word $w_t$ class, respectively. For given $t \in (0, \infty)$, we simply denote $s_T(w_t, T = t)$ as $s_T|_{T=t}$. A temperature $T$ with its best performance of final fit to RTs is denoted as $T^*$.

The extent to which a word's surprisal is affected by temperature scaling depends on the distribution and thus correlates with the entropy at word $w_t$. Consider an example of two five-class probability distributions $\boldsymbol{p}_i = [0.8, 0.05, 0.05, 0.05, 0.05]$ and $\boldsymbol{p}_j = [0.8, 0.2, 0, 0, 0]$, for which the word indicated by the first position in the probability vector has identical surprisal in both $\boldsymbol{p}_i$ and $\boldsymbol{p}_j$. Notably, $\boldsymbol{p}_i$ is more uniform and $\boldsymbol{p}_j$ is more peaked, resulting in distinct entropy characteristics: $H(w_i|\boldsymbol{w}_{<i}) > H(w_j|\boldsymbol{w}_{<j})$, where the entropy defined as the expectation of surprisal of current word $w_t$ over vocabulary, $H(w_t|\boldsymbol{w}_{<t}) = \mathbb{E}_{w' \sim p(\cdot|\boldsymbol{w}_{<t})}[s(w')] = -\sum_{w' \in \overline{\mathcal{W}}} p(w'|\boldsymbol{w}_{<t})\log_2 p(w'|\boldsymbol{w}_{<t})$, where $\overline{\mathcal{W}} = \mathcal{W} \cup \{EOS\}$ denotes the set of vocabulary $\mathcal{W}$ with EOS token. Fig. 1 illustrates **a greater increase in surprisal** for a word with a more uniform distribution than with a more peaked distribution. This figure also anecdotally shows that the effect of applying temperature scaling with $T > 1$ is similar to the effect of setting $\alpha < 1$ in Rényi entropy. We will discuss the relationship between these parameters in more detail in the next section.

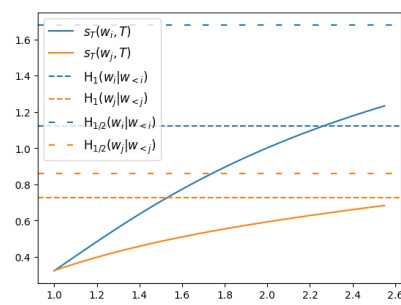

Figure 1: Temperature-scaled surprisal $s_T(w_t, T)$ with corresponding $T \in [1, 2.5]$ for two random five-class probability distributions: $p_i = [0.8, 0.05, 0.05, 0.05, 0.05]$ and $p_j = [0.8, 0.2, 0, 0, 0]$. Dashed lines show Shannon entropy ($H_1$). Loosely dashed lines show Rényi entropy with $\alpha = 1/2$ ($H_{1/2}$).

## 3.4 CONNECTION TO CONTEXTUAL RÉNYI ENTROPY

While a lot of work has investigated the effect of next word entropy on reading times (Hale, 2003; 2006; Linzen & Jaeger, 2014; Angele et al., 2015; van Schijndel & Linzen, 2019; Aurnhammer & Frank, 2019; Pimentel et al., 2023), we will here focus on contextual Rényi entropy (the entropy of the probability distribution at the current time stamp, which is parameterized by $\alpha$), as proposed in Pimentel et al. (2023) to represent human anticipatory reading process. Pimentel et al. (2023) find that Rényi entropy with an optimal $\alpha^*$ in the range of $(0, 1)$ (around $1/2$) obtains the best performance in reading time prediction (compared to Shannon Entropy ($\alpha = 1$) or compared to unscaled surprisal estimates).

Mathematically, Contextual Rényi entropy (Rényi, 1961) is defined as:

$$H_\alpha(w_t \mid \boldsymbol{w}_{<t}) = \lim_{\beta \to \alpha} \frac{1}{1-\beta}\log_2 \sum_{w \in \overline{\mathcal{W}}} (p(w|\boldsymbol{w}_{<t}))^\beta.\tag{7}$$

For given $\alpha^{'} \in (0, \infty)$, we simply denote $\mathrm{H}_\alpha(w_t \mid \boldsymbol{w}_{<t})|_{\alpha=\alpha'}$ as $\mathrm{H}_\alpha|_{\alpha=\alpha'}$.

**Theorem 1** (Monotonicity of $s_T(w_t, T)$ and $\mathrm{H}_\alpha(w_t \mid \boldsymbol{w}_{<t})$). *Given any probability distribution $\boldsymbol{p}$ with actual-word probability $p_{w_t} > 1/K$, where $K$ is the number of classes, temperature-scaled surprisal $s_T(w_t, T)$ is strictly monotonically increasing in $\Delta_T \in [1, \infty]$, Rényi entropy $\mathrm{H}_\alpha(w_t \mid \boldsymbol{w}_{<t})$ is strictly monotonically decreasing in $\Delta_\alpha \in [0, 1]$, especially,*

$$s_T|_{T=1} < s_T|_{T=T^*} < \lim_{T \to \infty} s_T(w_t, T) \tag{8}$$

$$\mathrm{H}_\alpha|_{\alpha=1} < \mathrm{H}_\alpha|_{\alpha=1/2} < \mathrm{H}_\alpha|_{\alpha=0}, \tag{9}$$

*where $T^*$ is the optimal $T$ of fit to RTs in the range of $\Delta_T$.*

**Theorem 2** *Rényi entropy with $\alpha = 0$ is equivalent to temperature-scaled surprisal with $T \to \infty$.*

$$\mathrm{H}_\alpha(w_t \mid \boldsymbol{w}_{<t})|_{\alpha=0} = \lim_{T \to \infty} s_T(w_t, T). \tag{10}$$

**Theorem 3** *For $K \geq 2$, the expectation of the L1 norm between Rényi entropy with $\alpha = 1$ and temperature-scaled surprisal with $T = 1$ has an upper bound.*

$$\mathbb{E}[|s_T|_{T=1} - \mathrm{H}_\alpha|_{\alpha=1}|] < \sqrt{\frac{1}{4}\log^2(K-1) + 1} \tag{11}$$

Proofs of the above theorems are shown in Appendix A. Theorem 2 claims the equivalence of temperature-scaled surprisal $s_T(w_t, T)$ and Rényi entropy $\mathrm{H}_\alpha$ when $T \to \infty$ and $\alpha = 0$. Theorem 3, on the other side, gives an upper bound when $T = 1$ and $\alpha = 1$. Intuitively, when $T \in (1, \infty)$, $s_T$ can be considered as a softened version of $s_T|_{T=1}$. Similarly, when $\alpha \in (0, 1)$, $\mathrm{H}_\alpha$ can be considered as a softened version of $\mathrm{H}_\alpha|_{\alpha=1}$. Mathematically, Theorem 1 provides the monotonicity of both functions within their respective domains. Hypothetically, given the above conditions, when tuning both functions with the aim of a better fit to RTs, $s_T|_{T=T^*}$ and $\mathrm{H}_\alpha|_{\alpha=1/2}$ might be close. Empirically, Fig. 2 illustrates the relationship between averaged Rényi entropy $\overline{\mathrm{H}}_\alpha|_{\alpha=\{0,1/2,1\}}$ and $\overline{s}_T|_{T=\{1,T^*,\infty\}}$ on probabilities on three corpora. Notably, $\overline{\mathrm{H}}_\alpha|_{\alpha=1/2}$ and $\overline{s}_T|_{T=T^*}$ are closely aligned, especially when compared with other entropy and surprisal

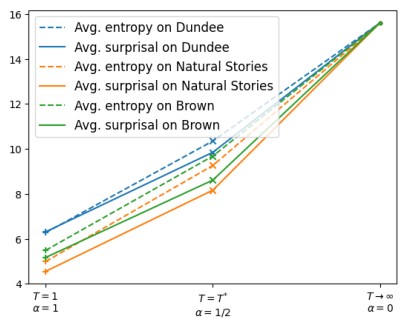

Figure 2: A comparison of averaged temperature-scaled surprisal $\overline{s}_T|_{T=\{1,T^*,\infty\}}$ and Rényi entropy $\overline{\mathrm{H}}_\alpha|_{\alpha=\{0,1/2,1\}}$.

data points. This empirical evidence partly verifies Theorem 2, Theorem 3 and our hypothesis.

## 4 EXPERIMENTAL SETUP

### 4.1 DATASETS

We conduct analyses on two self-paced reading corpora, the Natural Stories Corpus (Futrell et al., 2018) and the Brown Corpus (Smith & Levy, 2013), as well as on the Dundee Corpus (Kennedy et al., 2003) of eye-tracking measures. We follow previous work with respect to the preprocessing steps for each corpus (Kuribayashi et al., 2022; Shain et al., 2022). Appendix C shows details about the preprocessing steps of each corpus.

### 4.2 LANGUAGE MODELS

Recent observations showed that surprisal provided by LLMs with more parameters and lower perplexity is less predictive of self-paced reading times and eye-gaze durations (Shain et al., 2022; Oh & Schuler, 2023); across different experiments, GPT-2 (Radford et al., 2019) surprisals were found to predict human RTs the best. Therefore, we take four variants of pretrained GPT-2 (small, medium, large, xl) as our language models in all experiments. Following prior work, we obtain the surprisal for words composed of more than one subword by summing up the surprisal estimates of the subwords.

### 4.3 METRICS AND EVALUATION

We measure the predictive power of surprisal estimates from different language models, which is denoted as the log-likelihood difference per data point between a linear mixed-effects (LME) regression model with a predictor of surprisal estimates (target model) and a model without surprisal (base model), following Goodkind & Bicknell (2018); Wilcox et al. (2020). More specifically, the metric of delta log-likelihood is defined as:

$$\Delta_{\text{llh}} = \text{llh}(f_\phi(\mathbf{v}^{tgt})) - \text{llh}(f_\phi(\mathbf{v}^{base})), \tag{12}$$

where $\mathbf{v}^{tgt}$ is *target predictor variables* that include baseline predictor variables as well as predictor variables of our interest, such as surprisal or temperature-scaled surprisal. $\mathbf{v}^{base}$ is *base predictor variables* only including baseline predictor variables. The greater the value of $\Delta_{\text{llh}}$, the more valuable the additional surprisal estimates are for predicting human reading times.

For the calibration error evaluation, we set the number of bins $M$ to 15 for both ECE and CECE, aligning with prior literature, such as works by Guo et al. (2017); Kumar et al. (2019); Rahimi et al. (2020b), to ensure consistency in addressing problems where comparable probability ranges are relevant. The calibration metrics (ECE and CECE) are evaluated separately on each of the reading time corpus $\mathcal{D}$. For simplicity, our calibration evaluation is conducted at the token level. Given that many words have extremely low probabilities and thus are often grouped into a single bin, we also evaluate the calibration error *under the log probability binning scheme*. For other descriptions regarding the metrics and evaluation, see Appendix. G.

## 5 RESULTS

### 5.1 CALIBRATION OF LLMS

Table 1 shows ECE and CECE in log binning scheme for GPT-2 models of different sizes. **LLMs are in general well calibrated on language modeling.** Besides, **LLM calibration improves with scale.** Larger LMs are more calibrated. This conclusion is consistent with calibration investigation evaluated in BIG-bench multiple-choice tasks in Srivastava et al. (2023) as well as in several tasks including language modelling in Zhu et al. (2023).

### 5.2 MAIN RESULT: TEMPERATURE-SCALED SURPRISAL IMPROVES HUMAN READING TIME PREDICTION

We evaluate the predictive power of temperature-scaled surprisal. We scale $T$ in the range of $[1, 10]$ and measure $\Delta_{\text{llh}}$, see Fig 3. First, a confirmatory observation regarding the relationship between model size and predictive power: At $T = 1$, GPT-2 small exhibits the best predictive performance, and as the model size increases, $\Delta_{\text{llh}}$ declines, which is consistent with previous studies (Shain et al., 2022; Oh et al., 2022; Oh & Schuler, 2023). Secondly, **scaling the surprisal with $T > 1$ can significantly improve the predictive power across all corpora and LLMs.** With optimal $T^*$, on Dundee, Natural Stories, and Brown, the $\Delta_{\text{llh}}$ improvement is 23-43%, 60-89%, and 14-24%, respectively. We also observe a consistent pattern: when increasing $T$, $\Delta_{\text{llh}}$ first rises then declines; **the optimal value $T^*$ falls within the range of (2, 3) (around 2.5) across all models and corpora** in our setting. At $T^*$, even though the impact of model size on final performance is not fully recovered, the disparity diminishes. Smaller models continue to outperform, but the extent of model sizes influencing performance is reduced.

Finally, **larger LMs typically have a larger human-likeness calibration error**, shown in Table 1. Larger LMs also require a higher value of T to reach their best performance and have a greater increase by temperature-scaled surprisal.

### 5.3 CALIBRATION ERROR VS. RT PREDICTION ERROR

Table 2 shows ECE and CECE in both equally-spaced and log binning schemes when $T$ equals 1 and $T^*$ on three corpora. Probability distribution shaped by an optimal $T^*$ learnt for fit to human RTs drastically hurts the model calibration regarding these two metrics. ECE and CECE with $T^*$ are more than 10 times worse than values with $T = 1$. This discrepancy can be attributed to the

Table 1: Optimal $T^*$, $\Delta_{\text{llh}}$ improvement (%) ($\Delta_{\text{llh}}+ = (\Delta_{\text{llh}}(T = T^*) - \Delta_{\text{llh}}(T = 1))/\Delta_{\text{llh}}(T = 1)$), and calibration errors (HCE$_{\text{TS}}$, % ECE and % CECE) for GPT2s on Dundee, Natural Stories (NS) and Brown. $\Delta_{\text{llh}}$ values are multiplied by 1000. ECE and CECE are evaluated on log binning scheme.

|        |    | $T^*$ | $\Delta_{\text{llh}}$+ | HCE$_{\text{TS}}$ ↓ | ECE$_{\text{log}}$ ↓ | CECE$_{\text{log}}$ ↓ |
|--------|----|-------|------------------------|----------------------|----------------------|------------------------|
| Dundee | s  | 2.75  | 22.5                   | 3.11                 | 1.59                 | 4.07E-03               |
|        | m  | 3.0   | 42.0                   | 3.61                 | 1.74                 | 4.13E-03               |
|        | l  | 3.0   | 39.9                   | 3.82                 | 1.55                 | 3.99E-03               |
|        | xl | 3.25  | 43.2                   | 4.13                 | 1.29                 | 3.84E-03               |
| NS     | s  | 2.5   | 60.3                   | 3.31                 | 1.91                 | 1.53E-02               |
|        | m  | 2.5   | 63.0                   | 3.50                 | 1.80                 | 1.50E-02               |
|        | l  | 2.5   | 82.6                   | 3.97                 | 1.70                 | 1.40E-02               |
|        | xl | 2.5   | 89.0                   | 4.07                 | 1.56                 | 1.35E-02               |
| Brown  | s  | 2.5   | 13.7                   | 3.10                 | 1.69                 | 1.53E-02               |
|        | m  | 2.5   | 16.2                   | 3.29                 | 2.27                 | 1.51E-02               |
|        | l  | 2.75  | 21.8                   | 4.18                 | 1.58                 | 1.44E-02               |
|        | xl | 2.75  | 24.4                   | 4.29                 | 1.56                 | 1.38E-02               |

Table 2: Expected calibration errors (% ECE and % CECE) for GPT-2 small on Dundee, Natural Stories (NS) and Brown. Results are all evaluated on the equally-spaced binning scheme and log binning scheme.

|        | $T$     | ECE↓  | ECE$_{\text{log}}$ ↓ | CECE↓    | CECE$_{\text{log}}$ ↓ |
|--------|---------|-------|----------------------|----------|------------------------|
| Dundee | 1       | 1.43  | 1.59                 | 4.05E-03 | 4.07E-03               |
|        | $T^*$   | 28.68 | 28.68                | 7.30E-03 | 9.88E-03               |
| NS     | 1       | 2.48  | 1.91                 | 1.83E-02 | 1.53E-02               |
|        | $T^*$   | 35.85 | 35.85                | 3.16E-02 | 3.97E-02               |
| Brown  | 1       | 1.82  | 1.69                 | 1.67E-02 | 1.53E-02               |
|        | $T^*$   | 33.16 | 33.16                | 2.75E-02 | 3.34E-02               |

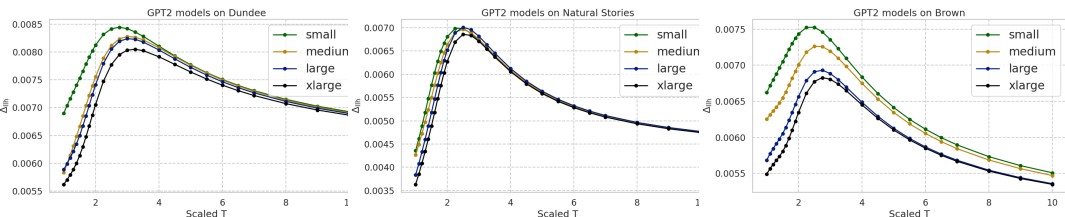

Figure 3: Relationship between $\Delta_{\text{llh}}$ of GPT-2 models and corresponding temperature. T is scaled from 1.0 to 10.

different minima of deviations in LM human RT prediction and expected calibration error. The former is minimized towards words where LMs surprisal significantly deviates from human processing difficulty, while the latter is typically minimized with respect to the negative log-likelihood on a hold-out dataset (Guo et al., 2017; Rahimi et al., 2020a).

## 6 LINGUISTIC ANALYSIS

Next we want to gain insight into what words benefit the most from temperature scaling. To this end, we analyze residuals from fitting LME regression models, identifying data points where scaling the temperature parameter notably enhances the fit of human RTs. Specifically, we quantify the improvement in fit by comparing the mean squared error (MSE) before and after adjusting the temperature to its optimal value as follows:

$$\Delta_{\text{MSE}}(F) = \text{MSE}_{T=1}(x_F) - \text{MSE}_{T=T^*}(x_F), \tag{13}$$

where $\text{MSE}_{T=T'}(x_F)$ is the MSE calculated by all the data $x_F$ under the linguistic factor $F$. The difference $\Delta_{\text{MSE}}(F)$ thus quantifies the impact of scaling relative to the linguistic factor $F$. A higher $\Delta_{\text{MSE}}(F)$ signifies a greater influence of temperature-scaled surprisal of factor $F$. To ensure sufficient data in each subset, we only consider subsets including more than 1% of the data in each corpus.

### 6.1 INFLUENCE OF LOW PROBABILITY WORDS

Given that temperature scaling enhances human likeness by shaping the probability distribution, it is natural to think about investigating whether there exists an inherent relationship between the distribution of probability and $\Delta_{\text{MSE}}$. Specifically, one might ask questions like if samples with low

probability gain more from temperature scaling or the other way around. We find that high surprisal words benefit more from temperature scaling than low surprisal words, across all corpora, see Fig 9 in Appendix H.

## 6.2 INFLUENCE OF WORD TYPES

We investigate the effects of word-level properties, which include:

**Named entities**. Research has substantiated that named entities (NEs) require increased reading time for humans since during the processing of such words (Damasio et al., 2004; Wang et al., 2013). Oh & Schuler (2023) showed that NEs are among the top two significant factors contributing to the discrepancies of large and small LMs across all corpus-by-LM combinations. Therefore, we were wondering whether the effect of temperature-scaling might be driven by NE. To test this, we automatically tagged NEs using a BERT base model (Devlin et al., 2019) fined-tuned for NER[2].

**Part-of-speech tags**. Similarly, previous research has argued that the poor fit of large LMs is primarily due to assigning too low surprisal estimates to open-class words like nouns and adjectives (Oh & Schuler, 2023). We POS-tagged the corpora using the NLTK toolkit (Bird et al., 2009) with the default Penn Treebank Tag set. In the following, we mainly focus on the four classes of open-class tags, as well as a subset of the whole closed-class tags (CC).

**Results**. The result, as shown in Table 6.2, shows primary factors responsible for the benefit of using $s_T(w_t, T)$ for each corpus-by-LM combination. The top three influential subsets for each corpus are underlined. Among all datasets and models, **named entities perform to be the most beneficial word-level attribute. In contrast, closed-class words profit least from temperature scaling.** Performance trends are consistent across different model variants on the same corpus.

|  | GPT2 | Avg. | Named entities | | POS tags | | | | |
|---|---|---|---|---|---|---|---|---|---|
|  |  |  | NE | non-NE | NN | ADJ | VERB | ADV | CC |
| Dundee | s | 26.3 | 87.0 | 23.4 | 33.8 | 100.5 | -2.0 | 2.6 | 10.4 |
|  | m | 41.7 | 152.3 | 36.4 | 57.0 | 123.3 | 7.8 | 27.6 | 16.4 |
|  | l | 40.1 | 158.2 | 34.5 | 56.3 | 126.5 | 4.8 | 19.2 | 14.0 |
|  | xl | 41.4 | 168.2 | 35.4 | 60.0 | 125.5 | 6.9 | 19.7 | 13.5 |
| NS | s | 105.7 | 186.8 | 104.6 | 148.7 | 152.5 | 122.0 | 49.0 | 77.1 |
|  | m | 108.5 | 155.9 | 107.9 | 145.3 | 152.0 | 130.1 | 60.8 | 80.8 |
|  | l | 127.7 | 151.6 | 127.3 | 175.6 | 158.6 | 152.9 | 74.8 | 94.3 |
|  | xl | 123.3 | 141.8 | 123.1 | 163.6 | 145.4 | 161.2 | 81.5 | 89.0 |
| Brown | s | 37.2 | 266.0 | 28.1 | 54.3 | -65.2 | 138.1 | 32.1 | 5.9 |
|  | m | 41.4 | 257.6 | 32.8 | 71.4 | -60.6 | 137.5 | 38.6 | 3.5 |
|  | l | 42.6 | 265.3 | 51.1 | 69.9 | -110.3 | 160.8 | 17.2 | 24.7 |
|  | xl | 54.8 | 282.3 | 45.8 | 90.5 | -90.2 | 151.3 | 32.2 | 20.0 |

Figure 4: $\Delta_{\text{MSE}}$ measurement on word-level properties of GPT-2 models on Dundee, Natural Stories (NS) and Brown. Top-3 on each corpus-by-LM are underlined.

We also measured empirically how often temperature scaling increased vs. decreased the surprisal estimate of a word. Our results show that for ca. 90% of words, surprisal estimates are increased through temperature scaling across all word classes. For the subset of named entities, a slightly smaller percentage exhibits increased surprisal estimates. For a full analysis across different corpora and models, see Table 3 in Appendix B. A further analysis reveals that the primary benefit of temperature-scaled surprisal arises from the increase of low surprisal values, particularly noticeable in the case of named entities, see Table 4 in Appendix B.

## 6.3 INFLUENCE OF MULTIPLE-TOKEN WORDS

A fact that is often ignored (but see Nair & Resnik, 2023) is that modern LLMs use subword tokenization. This means that long words may consist of several tokens. In this case, the probability of the complete word is calculated by multiplying the probabilities of the subword tokens (and the word's surprisal is correspondingly calculated by adding the surprisals of the subwords). While this may often not matter, whether a word is tokenized into a single subword or several subwords can make a remarkable difference when applying temperature scaling: imagine a long / difficult word which has a low probability (and correspondingly a high surprisal). If this word were to be represented as a single subword token, temperature scaling might have the effect that the probability of this word gets *increased* during temperature scaling, and its surprisal estimate is hence decreased at $T > 1$.

If, on the other hand, the same word were to be composed of two subword tokens based on the LM's subword vocabulary, one or both of the subword tokens can be expected to have a higher probability

---

[2]Link: https://huggingface.co/dslim/bert-base-NER

(than a hypothetical single subword token), and it is possible that during temperature scaling, the probabilities of the subword tokens would each be *decreased* at $T > 1$, such that the sum of the surprisals of the subword tokens would be much higher, compared to the word's surprisal estimate at $T = 1$.

To summarize, whether the surprisal of a certain word would increase or decrease after temperature scaling could depend on whether that word happens to be included in the subword token vocabulary or not. Distributions of surprisal for single vs. multiple token words before and after temperature scaling are provided in Figure 8 in Appendix H. In order to quantify to what extent subword tokenization affects surprisal estimates, we conducted several analyses.

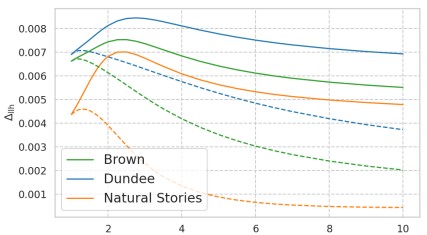

Fig. 5 shows $\Delta_{\text{llh}}$ under various conditions: scaling all words (consistent with experiments in Section 5.2) vs. taking into the analysis only the subset of single-token words. The comparison between the dashed and full curves highlights that **the benefit of temperature-scaled surprisal comes primarily from the scaling of multiple-token words**.

Figure 5: Relationship between $\Delta_{\text{llh}}$ of GPT-2 s on three corpora and corresponding temperature. T is scaled from 1.0 to 10. Dashed vs. full lines denote only scaling tokens in single-token words and scaling tokens in both single-token and multiple-token words, respectively.

Next, it is interesting to consider for what percentage of multiple-token words temperature-scaling *increases* the surprisal. We find that the **ratio of surprisal increase is higher for multi-token words than across single-token words** by ca. 6% on Dundee and Brown, see Table 7 in Appendix H for more details.

## 7 DISCUSSION

We found that temperature scaling is particularly effective for fitting RTs of named entities and open-class words; our later analysis on subword tokenization showed that this effect might be driven especially by words that are composed of several subword tokens. Of course open class words and named entities tend to be more complex and less likely to be contained in the vocabulary as single-word tokens, hence the observed correlation is to be expected.

So what does all of this mean for surprisal estimates from LLMs and reading time prediction? Firstly, it is possible that indeed the effect is driven by humans failing to accurately estimate the probability of rare words, because they do not reach sufficient language experience or because human language models do not track these probabilities well, in line with Oh & Schuler (2023); in this case, temperature-scaling rare words to which the LLM assigns a high probability (and hence a low surprisal) would be a good strategy to counteract the discrepancy between humans and LLMs.

Secondly, it is possible that the beneficial effect of temperature scaling is an artifact of subword tokenization, and that it would disappear if all words were composed of only a single subword token. In order to test this hypothesis, one would have to re-train a GPT-2 model from scratch using a vocabulary that at least includes all words that are contained in the reading time corpora, and then re-running the analysis to check whether a beneficial effect of temperature scaling can still be found.

Finally, it is also possible that the splitting of a word into subwords coincides with the reader fixating a word several times, and that these added fixations lead to an overestimate in RTs compared to the actual surprisal experienced by a human reader. Future work could investigate this hypothesis by analysing RTs on subwords instead of aggregated words. This would require re-calculating reading measures, and comes at the caveat that subword tokens are not cognitively plausible units.

## 8 CONCLUSION

This paper studies the prediction of human RTs from the perspective of probability distribution. We make the following contributions: (1) We demonstrate that the prediction of RTs can be significantly improved via temperature scaling of LLM probability estimates. (2) We establish that

temperature-scaled surprisal is related to Rényi entropy. (3) We demonstrate that the primary benefit of temperature-scaled surprisal is driven by words composed of several subword tokens. These words also tend to be rarer / long open-class words. Future work should investigate the interaction of subword tokenization and temperature scaling, as well as the issue of tokenization in the analysis of eye-tracking data.

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

## A   PROOF OF THEOREMS 1, 2 AND 3

**Theorem 1** (Monotonicity of $s_T(w_t, T)$ and $\mathrm{H}_\alpha(w_t \mid \boldsymbol{w}_{<t})$). *Given any probability distribution $\boldsymbol{p}$ with actual-word probability $p_{w_t} > 1/K$, where $K$ is the number of classes, temperature-scaled surprisal $s_T(w_t, T)$ is strictly monotonically increasing in $\Delta_T \in [1, \infty]$, Rényi entropy $\mathrm{H}_\alpha(w_t \mid \boldsymbol{w}_{<t})$ is strictly monotonically decreasing in $\Delta_\alpha \in [0, 1]$, especially,*

$$s_T|_{T=1} < s_T|_{T=T^*} < \lim_{T \to \infty} s_T(w_t, T) \tag{14}$$

$$\mathrm{H}_\alpha|_{\alpha=1} < \mathrm{H}_\alpha|_{\alpha=1/2} < \mathrm{H}_\alpha|_{\alpha=0}, \tag{15}$$

*where $T^*$ is the optimal $T$ in the range of $\Delta_T$.*

*Proof.* Eq. equation 8 can be easily verified by considering the monotonicity of temperature-scaled softmax output $\sigma_{SM}(\hat{\boldsymbol{z}}_{w_t}/T)$. The second part of Eq. equation 9 can be rewritten as:

$$\mathrm{H}_\alpha|_{\alpha=1/2} = 2 \log_2 \sum_{w \in \overline{\mathcal{W}}} \sqrt{p(w|\boldsymbol{w}_{<t})} \tag{16}$$

$$< 2 \log_2 \sqrt{K \sum_{w \in \overline{\mathcal{W}}} p(w|\boldsymbol{w}_{<t})} \tag{17}$$

$$= -\log_2(1/K) = \mathrm{H}_\alpha|_{\alpha=0}, \tag{18}$$

where for the step from Eq. equation 16 to Eq. equation 17 we use AM-QM inequality and $K$ is the number of classes in tokenizer. The first part of Eq. equation 9 can be rewritten as:

$$\mathrm{H}_\alpha|_{\alpha=1/2} = 2 \log_2 \sum_{w \in \overline{\mathcal{W}}} \sqrt{p(w|\boldsymbol{w}_{<t})} \tag{19}$$

$$> 2 \log_2 \sqrt{\prod_{w \in \overline{\mathcal{W}}} \left(\frac{1}{p(w|\boldsymbol{w}_{<t})}\right)^{p(w|\boldsymbol{w}_{<t})}} \tag{20}$$

$$= \sum_{w \in \overline{\mathcal{W}}} p(w|\boldsymbol{w}_{<t}) \log_2 p(w|\boldsymbol{w}_{<t}) = \mathrm{H}_\alpha|_{\alpha=1}, \tag{21}$$

where from Eq. equation 19 to Eq. equation 20 we use AM-GM inequality.

**Theorem 2** *Rényi entropy with $\alpha = 0$ is equivalent to temperature-scaled surprisal with $T \to \infty$.*

$$\mathrm{H}_\alpha(w_t \mid \boldsymbol{w}_{<t})|_{\alpha=0} = \lim_{T \to \infty} s_T(w_t, T). \tag{22}$$

*Proof.* By plugging in $\alpha = 0$, Contextual Rényi entropy recovers to be the entropy that readers concentrate on the count of potential words with nonzero probabilities, which is defined in Eq. (5) in Pimentel et al. (2023). As $T \to \infty$, temperature-scaled surprisal converges to the surprisal induced by random guessing. Given the assumtion that $p(w|\boldsymbol{w}_{<t}) > 0$ for each word $w \in \overline{\mathcal{W}}$, LHS becomes:

$$LHS = -\log_2(1/K), \tag{23}$$

where $K$ is the number of classes. As $T \to \infty$, RHS becomes:

$$RHS = -\lim_{T \to \infty} \log_2 \frac{e^{z_{w_t}/T}}{\sum_{w \in \overline{\mathcal{W}}} e^{z_w/T}} \tag{24}$$

$$= -\log_2(1/K) \tag{25}$$

**Theorem 3** *For $K \geq 2$, the expectation of the L1 norm between Rényi entropy with $\alpha = 1$ and temperature-scaled surprisal with $T = 1$ has an upper bound.*

$$\mathbb{E}[|s_T|_{T=1} - H_\alpha|_{\alpha=1}|] < \sqrt{\frac{1}{4} \log^2(K-1) + 1} \tag{26}$$

*Proof.* With Jensen's inequality, we have:

$$\mathbb{E}[|s_T|_{T=1} - H_\alpha|_{\alpha=1}|] \tag{27}$$

$$\leq \sqrt{\mathbb{E}[(s_T|_{T=1} - H_\alpha|_{\alpha=1})^2]} \tag{28}$$

$$= \sqrt{\mathbb{E}[(-\log_2 p_{w_t} - \sum_{w \in \overline{\mathcal{W}}} p(w)(-\log_2 p(w)))^2]} \tag{29}$$

$$= \sqrt{\mathrm{Var}[s_T|_{T=1}]} \tag{30}$$

$$< \sqrt{\frac{1}{4}\log^2(K-1) + 1}, \tag{31}$$

where $\mathrm{Var}[\cdot]$ denotes the variance. The last inequality is shown by Lemma 4, completing the proof of this theorem.

**Lemma 4** (Maximum variance of the surprisal). (See Theorem 8 and Lemma 15 in (Reeb & Wolf, 2015)). *Let $\rho = \mathrm{diag}(p_1, p_2, ..., p_d)$ be a state on a $d$-dimensional system. Let $-\log p_i$ be the surprisal of the output $i$ in this system. Define $N_d$ to be:*

$$N_d := \frac{1}{4}\log^2(d-1) + 1. \tag{32}$$

*For $d \geq 2$, the variance of surprisal has a tight upper bound:*

$$\mathrm{var}_\rho(-\log\rho) < N_d \tag{33}$$

## B  FURTHER ANALYSIS IN SECTION 6.2

We observe that **larger LMs exhibit an increased $\Delta_{\mathbf{MSE}}$** by utilizing temperature-scaled surprisal, as shown in the average column (Avg.) of Table 6.2. Specifically, on Dundee, the top 2 models achieving the largest improvement through temperature scaling are GPT-2 medium and xl, while GPT-2 large and xl have the most benefit on Natural Stories and Brown. This result is consistent with previously observed $\Delta_{\mathrm{llh}}$ improvement ($\Delta_{\mathrm{llh}}+$) across the corpus-by-LM reported in Table 1, **suggesting a correlation between model likelihood and MSEs of the regression models.** We do not observe a mismatch between them, as posited by Oh & Schuler (2023) that LME models achieve similar MSEs irrespective of obvious differences in model likelihood.

Regarding the effect of the change (increase or decrease) of actual-word probability on the final fit to RTs, we first analyzed the ratio of probabilities decreasing (or increasing) for all words, as well as for subsets with specific word-level properties, choosing named entities as the representative, as shown in Table 3. We observed that **probabilities of the majority of words (around 80-90%) decrease by temperature scaling**. Compared with the average across all word types (as indicated in the 'Avg.' column), named entities exhibit a lower ratio of probability reduction. Larger LMs tend to have a higher ratio, especially the ratio for named entities, likely because smaller models may lack the specific knowledge of less common terms, such as named entities. We further investigate the benefit of temperature-scaled surprisal (quantified by $\Delta_{\mathrm{MSE}}$) given the subset of words whose probability decreases (or increases). The results are in Table 4. On Dundee, the main gain arises from the reduction of large probabilities via temperature scaling. Conversely, for Natural Stories, the primary benefit comes more strongly from words with originally very low probability, which get more probable. For Brown, the effects are evenly split. This variation aligns with our theoretical intuition that **temperature scaling enhances the fit performance by making probabilities more smoothing and uncertain**, which means not only making high probabilities lower but also making super low probabilities higher and close to $1/K$, since a super low probability also means the model is confident in the incorrectness of certain classes. For named entities, the story is converse on Dundee vs. on Natural Stories and Brown, where for the latter two corpora, the advantage is primarily due to reducing the probabilities of highly predictable entities. We shed light to the possible reason of such a discrepancy in Fig 6, which displays the top 15 frequent words for GPT-2 small on each corpus. Notably, Natural Stories and Brown show a marked lack of words with increased probabilities (blue bins) compared to Dundee. This lack weakens the overall impact of rising probabilities (denoted by $\Delta_{\mathrm{MSE}}(p_{w_t}\uparrow)$). Specifically, on Brown, only 4 out of 15 top frequent words have the composition of increased probabilities, correlating with the largest discrepancy in $\Delta_{\mathrm{MSE}}$ between probabilities that decreased (329.7) and those that increased (-170.6) in Table 4.

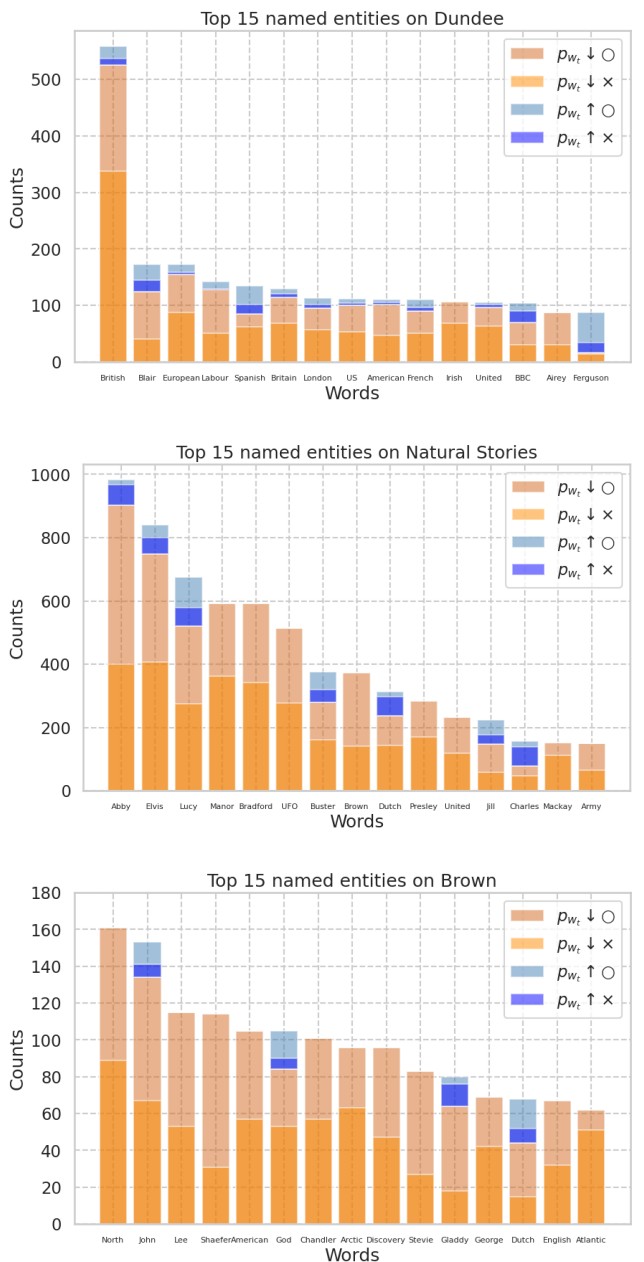

Figure 6: Top 15 frequent named entities for GPT-2 small on Dundee, Natural Stories and Brown. ↑ and ↓ denote being higher and smaller, respectively. ◯ and × denote unbeneficial words (absolute residual error increases) and beneficial words (absolute residual error decreases) by temperature scaling, respectively.

| Corpus | GPT2 | Avg. $p_{w_t}\downarrow$ | Avg. $|res|\downarrow$ | Named entities $p_{w_t}\downarrow$ | Named entities $|res|\downarrow$ |
|---|---|---|---|---|---|
| Dundee | s | 88.0 | 51.8 | 78.1 | 52.3 |
| | m | 89.6 | 52.5 | 80.1 | 54.1 |
| | l | 90.2 | 52.3 | 80.1 | 53.5 |
| | xl | 91.4 | 52.4 | 82.7 | 54.3 |
| Natural Stories | s | 93.8 | 55.0 | 85.3 | 51.8 |
| | m | 94.7 | 55.2 | 89.1 | 53.2 |
| | l | 93.5 | 55.7 | 89.1 | 53.4 |
| | xl | 92.1 | 55.5 | 88.2 | 52.8 |
| Brown | s | 91.8 | 51.5 | 87.3 | 50.9 |
| | m | 93.2 | 51.5 | 86.1 | 50.9 |
| | l | 93.3 | 51.8 | 88.6 | 52.1 |
| | xl | 93.5 | 51.7 | 87.8 | 53.3 |

Table 3: The ratio of probability of predicted word $p_{w_t}$ and the absolute value of residuals $|res|$ getting smaller for GPT-2 models on three corpora.

| Corpus | GPT2 | Avg. $p_{w_t}\downarrow$ | Avg. $p_{w_t}\uparrow$ | NE $p_{w_t}\downarrow$ | NE $p_{w_t}\uparrow^*$ | non-NE $p_{w_t}\downarrow$ | non-NE $p_{w_t}\uparrow$ |
|---|---|---|---|---|---|---|---|
| Dundee | s | **27.4** | 18.2 | 81.3 | **107.2** | **25.1** | 10.1 |
| | m | **41.9** | 39.8 | 139.1 | **205.6** | **37.8** | 23.9 |
| | l | **41.0** | 31.3 | 156.1 | **166.6** | **36.2** | 18.0 |
| | xl | **42.5** | 29.8 | **170.2** | 158.8 | **37.0** | 16.9 |
| Natural Stories | s | 94.5 | **275.6** | **218.5** | 3.0 | 92.9 | **284.9** |
| | m | 105.7 | **158.3** | **179.3** | -34.9 | 104.7 | **163.9** |
| | l | 125 | **166.1** | **197.5** | -224.8 | 124 | **175.4** |
| | xl | 121.8 | **140.7** | **197.3** | -272.6 | 120.8 | **149.5** |
| Brown | s | **37.6** | 32.6 | **329.7** | -170.6 | 26.6 | **45.5** |
| | m | 39.1 | **72.3** | 276 | 143.6 | 30.5 | **66.3** |
| | l | **52.7** | 28.1 | **325.8** | -205.9 | 42.5 | **44.4** |
| | xl | 50.9 | **111.5** | **298.2** | 168.2 | 41.7 | **107.1** |

Table 4: Given words whose probability decreases (and increases), the corresponding $\Delta_{\mathrm{MSE}}(p_{w_t}\downarrow)$ (and $\Delta_{\mathrm{MSE}}(p_{w_t}\uparrow)$) measurement for GPT-2 models on Dundee, Natural Stories (NS) and Brown. A higher $\Delta_{\mathrm{MSE}}$ is displayed in bold in the average across all word types (Avg.), named entities (NE), and non-named entities (non-NE) columns, respectively, for each corpus-by-LM combination. The column with $*$ indicates insufficient (less than 1%) data.

## C  PROPROCESSING STEPS

On Dundee ET corpus (Kennedy et al., 2003), we use the first-pass gaze duration. Following prior work (Kuribayashi et al., 2022), we remove words containing numbers or punctuation, words that are either the first or the last one in a line, as well as words whose previous words contain numbers or punctuation. On Natural Stories SPR corpus (Futrell et al., 2018), following Shain et al. (2022), we remove words if the RT is less than 100ms or greater than 3,000ms, if the words are in the first or last position of each story, if participants answered less than 5 out of 8 comprehension questions correctly, if words contain numbers or punctuation, and if words whose previous words containing numbers or punctuation. On Brown SPR corpus (Smith & Levy, 2013), following Shain et al. (2022), we remove words if the RT is less than 100ms or greater than 3,000ms and if words contain numbers or punctuation.

## D  EXPLORING FURTHER EFFECTIVENESS OF TEMPERATURE-SCALED SURPRISAL OVER BASIC PREDICTORS

In this section, we explore the question of whether the benefit of temperature-scaled surprisal holds only for regression models already containing other predictors such as length and frequency. We

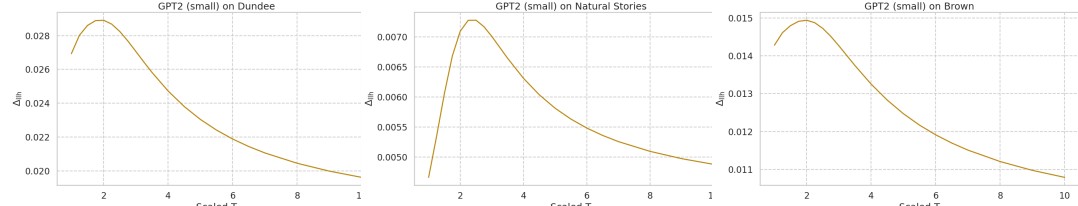

Figure 7: Relationship between $\Delta_{\text{llh}}$ of GPT-2 small and corresponding temperature. T is scaled from 1.0 to 10. Base predictor variables $\mathbf{v}^{base}$ and target predictor variables are 0 and temperature-scaled surprisal $s_T(w_t, T)$, respectively.

| | | $\text{ECE}_{\text{single}}$ | $\text{ECE}_{\text{multiple}}$ |
|---|---|---|---|
| Dundee | $T = 1$ | 1.98 | 2.05 |
| | $T = T*$ | 25.58 | 36.10 |
| Natural Stories | $T = 1$ | 2.20 | 3.78 |
| | $T = T*$ | 32.38 | 47.02 |
| Brown | $T = 1$ | 1.69 | 3.86 |
| | $T = T*$ | 28.70 | 42.99 |

Table 5: Expected calibration errors of tokens in single-token (% $\text{ECE}_{\text{single}}$) and multiple-token words (% $\text{ECE}_{\text{multiple}}$) before and after temperature scaling for GPT-2 small on Dundee, Natural Stories and Brown. Results are all evaluated on the equally-spaced binning scheme.

conduct experiments similar to those detailed in Section 5.2 while setting base predictor variables $\mathbf{v}^{base}$ to 0 and target predictor variables $\mathbf{v}^{tgt}$ to only temperature-scaled surprisal $s_T(w_t, T)$ in Eq. 12. Fig. 7 shows that **excluding base predictors decrease but not totally impact the effectiveness of temperature-scaled surprisal.**

## E    CALIBRATION ERROR FOR SINGLE-TOKEN AND MULTIPLE-TOKEN WORDS

In Table 5, we demonstrate the calibration error (% ECE) for single-token and multiple-token words for GPT-2 small. Calibration evaluation is conducted at the token level as before. Results indicate that **multiple-token words show larger calibration errors than single-token words.**

## F    PROBABILITY DISTRIBUTION BEFORE AND AFTER TEMPERATURE SCALING

Fig. 8 shows actual-word probability distribution before and after temperature scaling for GPT-2 small on three corpora. **Multiple-token words tend to have smaller probabilities than single-token words**, both before and after temperature scaling.

## G    OTHER DESCRIPTIONS ON METRICS AND EVALUATION

We evaluate calibration error (% ECE and % CECE) in both equally-spaced and log binning schemes. In equally-spaced binning scheme, the samples are grouped into $M \in \mathbb{N}$ equally-spaced interval bins based on their confidences $\hat{p}_i$. Conversely, the log binning scheme operates under an *empirical upper limit* for $-\log_2 \hat{p}_i$, denoted as $\max(-\log_2 \hat{p})$. Table 6 shows ranges of $\hat{p}$ and $-\log_2 \hat{p}$ for GPT2s on three corpora. For this scheme, we establish $M \in \mathbb{N}$ log-equally-spaced interval bins within the range of $(0, \max(-\log_2 \hat{p})]$.

We investigate scaling $T \in [1, 10]$, considering both densely and sparsely distributed points. The values examined are detailed as follows: [1.0, 1.1, ..., 1.9] for dense intervals, [2.0, 2.25, ..., 3.25] for moderately spaced intervals, and [3.5, 4.0, ..., 10.0] for sparse intervals.

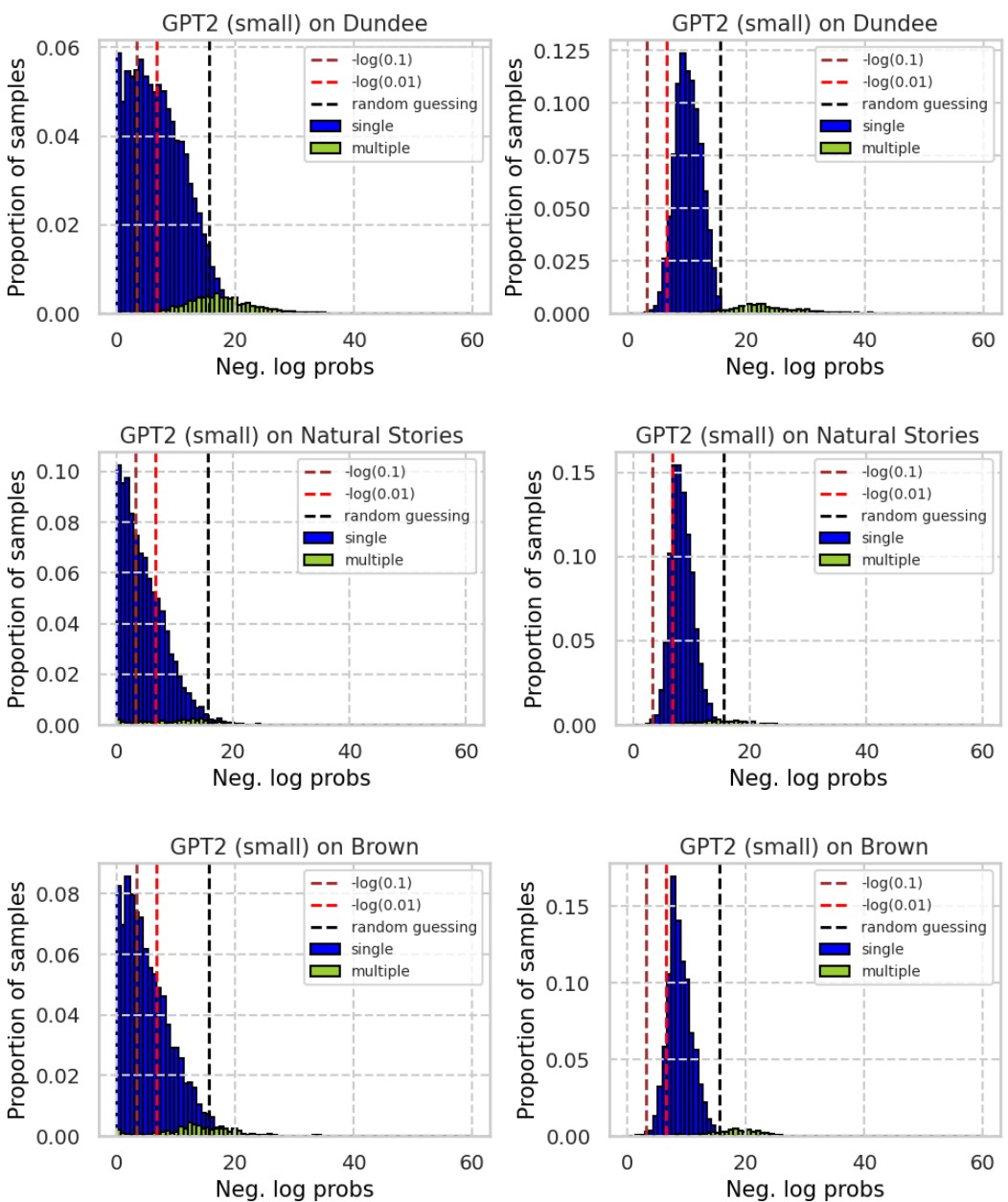

Figure 8: Distribution of negative log actual-word probability (surprisal) before (left side of figure) and after (right side of figure) temperature scaling for single-token and multiple-token words for GPT-2 small on three corpora. Values of surprisal with probability of 0.1, 0.01 and 1/K (random guessing) are displayed using dash lines.

|  | $\hat{p}$ | $-\log_2 \hat{p}$ |
|---|---|---|
| Dundee | [4.99e-03, 1) | (0, 7.65] |
| Natural Stories | [8.567e-03, 1) | (0, 6.87] |
| Brown | [8.15e-03, 1) | (0, 6.94] |

Table 6: Ranges of $\hat{p}$ and $-\log_2 \hat{p}$ for GPT2s on Dundee, Natural Stories and Brown.

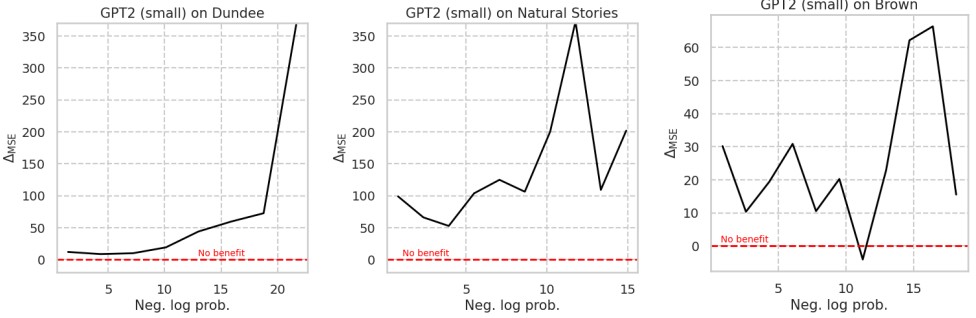

Figure 9: Relationship between $\Delta_{\mathrm{MSE}}$ and negative log actual-word probability (surprisal). We take the number of bins to 20. Red dashed lines denote $\Delta_{\mathrm{MSE}} = 0$. Subsets containing less than 1% of data are ignored for each corpus.

Following Kuribayashi et al. (2022), reading times of a base model are modelled by the following formula:

$$rt \sim \text{freq} * \text{length} + \text{freq\_prev\_1} * \text{length\_prev\_1} + (1|\text{article}) + (1|\text{subj\_id}) \tag{34}$$

A target model additionally includes surprisal estimates of current words and previous words, [surprisal, surprisal_prev_1, surprisal_prev_2]. On Dundee corpus, both models also include features [screenN, lineN, segmentN].

## H    OTHER RESULTS IN SECTION 6

|  | ratio of $p_{w_t}\downarrow$ | | | | ratio of named entities | | | |
|---|---|---|---|---|---|---|---|---|
|  | # = 1 | #>1 | # = 2 | # = 3 | # = 1 | #>1 | # = 2 | # = 3 |
| Dundee | 87.6 | 93.7 | 90.6 | 98.3 | 3.7 | 16.3 | 16.6 | 17.4 |
| Natural Stories | 92.1 | 93.0 | 92.2 | 97.2* | 1.3 | 3.5 | 3.3 | 4.7* |
| Brown | 93.0 | 98.1 | 97.6 | 35.2* | 3.3 | 12.3 | 10.9 | 17.0* |

Table 7: This table displays the ratio of words with decreasing probability ($p_{w_t}\downarrow$) and the ratio of named entities on subsets for both single-token words (#=1) and multiple-token words (#¿1) for GPT-2 small on three corpora. Numbers marked with $*$ indicate subsets with insufficient (less than 1%) data.

| | #=1 | | #>1 | | #=2 | | #=3 | |
|---|---|---|---|---|---|---|---|---|
| | $p_{w_t}\downarrow$ | $p_{w_t}\uparrow$ | $p_{w_t}\downarrow$ | $p_{w_t}\uparrow$ | $p_{w_t}\downarrow$ | $p_{w_t}\uparrow$ | $p_{w_t}\downarrow$ | $p_{w_t}\uparrow$ |
| Dundee | 8.0 | 19.6 | 269.5 | -20.3* | 50.5 | 26.6* | 497.4 | 125.4** |
| NS | 117.3 | 142.3 | 242.5 | 93.0* | 312.6 | 95.8* | -123.9* | 50.6** |
| Brown | 35.2 | -61 | 327.3 | 5290.2** | 17.3 | 5290.2** | 655* | 0** |

Table 8: Given words with decreasing (and increasing) probability, the corresponding $\Delta_{\mathrm{MSE}}(p_{w_t}\downarrow)$ (and $\Delta_{\mathrm{MSE}}(p_{w_t}\uparrow)$) measurement for both single-token words (#=1) and multiple-token words (#¿1) for GPT-2 small on three corpora. Numbers marked with $*$ indicate subsets with insufficient (less than 1%) data. Numbers marked with $**$ indicate subsets with super insufficient (around or less than 0.1%) data.

