# OpenReview forum: "TEMPERATURE-SCALING SURPRISAL ESTIMATES IMPROVE FIT TO HUMAN READING TIMES – BUT DOES IT DO SO FOR THE “RIGHT REASONS”?"
_ICLR.cc/2024/Workshop/Re-Align — ICLR 2024 Workshop Re-Align Poster_

### Official Review · Reviewer_kwem · 2024-02-22
**Interesting to see a new method to improve LLM fit to human reading times**

**Rating:** 2
**Fit:** 3
**Confidence:** 1

**Workshop Review:**

Summary:
- This paper studies using AI large language models (LLMs) to predict human reading times (RTs). This is motivated by investigating the next-word prediction probability estimates of LLMs compared to humans. The paper shows that human RT prediction can be improved using temperature scaling, i.e., making the probability distribution more uniform.
- The paper runs further experiments, showing that this improvement is primarily driven by words composed of several subword tokens and also, named entities and open-class words, e.g., nouns and adjectives.

Strengths:
1. Tested on multiple sizes of LLM (gpt-2: small, medium, large, xl)
2. Tested on multiple human RT datasets (Natural Stories, Brown, Dundee)

Weaknesses:
1. As I was reading the paper, I kept thinking of one hypothesis for why temperature-scaling improves LLM prediction of human RTs: LLM surprisals in normal conditions poorly match human RTs, but temperature-scaling makes the LLM surprisals more uniform, hence matching human RTs better. This is supported by some of their other results, including the result that larger LLMs generally have worse delta-llh, and also require a higher value of T to reach their best performance (Table 1). This is somewhat related to Paragraph 2 of the paper's Discussion section, but I don't think they address this hypothesis directly.
2. I hope the paper can explain a bit more about why it matters to find new methods (e.g., temperature-scaling) to improve LLM prediction of human RT, i.e., what would achieving this mean? Also, why the paper's results are important -- knowing that temperature-scaling surprisal improvement is driven by multi-subword words and open-class words, how does this change our understanding of LLMs or human reading?

Minor notes:
1. The paper cites a paper saying that LLM probability estimates are often overconfident compared to ground-truth. I wondered if humans are also overconfident compared to ground-truth. Knowing this would help me understand why temperature scaling improves LLM fit to human RTs.
2. The paper describes "LLMs". However, GPT-2 is not really "large" (except the xl version they tested, maybe) compared to much newer LLM models.
3. Can the paper provide an intuitive visualization and explanation of how well each/any of these LLMs predict RTs? I think it would aid understanding. How close are LLMs to perfectly predicting human RT?
4. The paper title asks: BUT DOES IT DO SO FOR THE “RIGHT REASONS”? I wish the paper discusses more about what they think are the "right reasons", and also answers the question (whether temperature-scaling estimates improve fit to human RTs for the right reasons).

**Reason For Not Giving Higher Score:**

N/A

**Reason For Not Giving Lower Score:**

N/A

**Reviewer Domain:**

machine learning

---

### Official Review · Reviewer_pYcy · 2024-02-22
**Well-crafted paper with opportunity to promote discussion**

**Rating:** 3
**Fit:** 3
**Confidence:** 3

**Workshop Review:**

### Summary
- It has been described in the literature that surprisal estimates from larger LLMs begin to reduce in fit to human RTs. The authors propose that this is due to the lowest perplexity models becoming the most "peaky" and propose using temperature rescaling to smooth out the resulting distributions. This has the intended effect of improving RT fit for large models. The authors investigate precisely on what subset of RT data this improvement is most pronounced and highlight interesting pathologies of subword tokenization.

### Clarity
Strengths:
- This is a clearly written manuscript. The authors make ample effort to discuss existing literature, formulate the research question and contribution, and outline the technical background.

Weaknesses:
- While the writing is clear, several of the figures could use work. Also, please label the y-axes in figs 1 and 2.
- Please include the full specification of the LMEs fixed and random effects structure in 4.3

### Correctness
Strengths:
- The authors are careful to state their assumptions and the implications of their results. The investigation into exactly which subset of the RT data these fit improvements are most pronounced is excellent.

### Novelty & Interest to the community
Strength:
- This is a clear investigation of how temperature rescaling can be used to adjust model probability estimates toward human distributions. I think the workshop will find this interesting both within psycholinguistics and more broadly. The results on some of these subword tokenization pathologies are particularly worthy of discussion.

Weakness:
- While an extremely well-constructed paper, the core idea of temperature rescaling to improve calibration between two distributions is quite trivial (and has been used in other domains although not for this particular LM surprisal - human RT calibration problem).

**Reason For Not Giving Higher Score:**

N/A

**Reason For Not Giving Lower Score:**

This is a clear and well-constructed paper whose results are impactful in the subfield of sentence processing in psycholinguistics, but whose broader themes are relevant for numerous discussions comparing measurements from humans and probabilistic models.

**Reviewer Domain:**

cognitive science

---

### Official Review · Reviewer_burx · 2024-02-23
**Word representations, reading time, and language model calibration**

**Rating:** 2
**Fit:** 3
**Confidence:** 2

**Workshop Review:**

Why do humans spend longer reading some words than others? Work in psycholinguistics has long emphasized the importance of word surprisal as a driving factor; yet, such measures require an estimate of the underlying probability of a word. Language models are revising our collective understanding of word probabilities; while the authors note that one may expect that more faithful estimates of the probabilities of words in our world would lead to better fits to human surprisal judgments, it is not clear that they do. The authors try to figure out why in this work.

In particular, the authors study whether the calibration of language models' probability distributions underlies such discrepancies. Through a series of experiments with GPT-2 variants, the authors find that indeed, modulating language models' distributions through temperature scaling (specifically, flattening the distributions) yields better fits to human reading times. They conduct a more granular investigation to identify differences at individual word levels. This work nicely informs a better understanding of human and machine representations of words, and uncertainty over these representations; the work is a good topical fit to this workshop.

While I appreciated that the authors provided background on the metrics they considered (Section 3 was well-written), unfortunately I found the bulk of the paper to be quite clunky to read. The graphs were also quite small and not easily interpretable; there was also no sense of error bars in either tables or figures (which I would strongly encourage the authors to add in a future version). Coupled with my confusion from the text, this made it hard to deeply understand and assess the results.

Additionally, I take some issue with the way the authors discuss humans throughout the piece. I would encourage the authors to avoid the phrase "human language model" or clarify a bit more what they mean there... I assume this refers to humans' internal model(s) of language processing? The passage: “Temperature tuning inherently shifts the model’s output in the generation’s quality/diversity spectrum (Caccia et al., 2018), with higher temperature decreasing the quality of generation while improving its diversity. This also aligns with our intuition: increasing the predictive diversity of surprisal provided by LLMs to be more human-like" implies that humans have poor quality language generation...? Yet, humans are great language generators overall... and can be much better than the GPT-2 model considered by the authors here. I think more nuance here would be beneficial.

For future experiments, I would be keen to see how the authors' results stand up for larger language models, like LLaMA or Mistral. I believe the experiments should be doable with the models' logprobs?

I will caveat my review that I am not well-versed in either psycholinguistics literature or the relationship between language model and human surprisal (which seems to be an active area of study), so I have a hard time assessing novelty. However, I did find the paper interesting and imagine others at the workshop would as well. I think the paper could spark discussion, and as such, I recommend to accept.

**Reason For Not Giving Higher Score:**

My biggest challenge was lack of clarity. As mentioned, I had trouble to understand large parts of the text as well as the figures, which made it hard to interpret the results faithfully.

**Reason For Not Giving Lower Score:**

I believe the work will nicely appeal to the representation alignment community, straddling both ML and psycholinguistics. The work appears timely, and I think can spark conversation.

**Reviewer Domain:**

machine learning

---

### Decision · Program_Chairs · 2024-03-02

Accept (Poster)